# Recent Advances in Microbial Synthesis of Poly-γ-Glutamic Acid: A Review

**DOI:** 10.3390/foods11050739

**Published:** 2022-03-02

**Authors:** Danfeng Li, Lizhen Hou, Yaxin Gao, Zhiliang Tian, Bei Fan, Fengzhong Wang, Shuying Li

**Affiliations:** 1Institute of Food Science and Technology, Chinese Academy of Agricultural Sciences, No. 2 Yuan Ming Yuan West Road, Beijing 100193, China; 82101205002@caas.cn (D.L.); houlizhen2021@sina.com (L.H.); gaoyx2021@sina.com (Y.G.); 82101202159@caas.cn (Z.T.); fanbei@caas.cn (B.F.); 2Key Laboratory of Agro-Products Quality and Safety Control in Storage and Transport Process, Ministry of Agriculture and Rural Affairs, Chinese Academy of Agricultural Sciences, Beijing 100193, China; 3Key Laboratory of Agro-Products Processing, Ministry of Agriculture and Rural Affairs, Chinese Academy of Agricultural Sciences, Beijing 100193, China

**Keywords:** poly-γ-glutamic acid, *Bacillus* species, microbial synthesis, metabolic pathway, industrial application

## Abstract

Poly-γ-glutamic acid (γ-PGA) is a natural, safe, non-immunogenic, biodegradable, and environmentally friendly glutamic biopolymer. γ-PGA has been regarded as a promising bio-based materials in the food field, medical field, even in environmental engineering field, and other industrial fields. Microbial synthesis is an economical and effective way to synthesize γ-PGA. *Bacillus* species are the most widely studied producing strains. γ-PGA biosynthesis involves metabolic pathway of racemization, polymerization, transfer, and catabolism. Although microbial synthesis of γ-PGA has already been used extensively, productivity and yield remain the major constraints for its industrial application. Metabolic regulation is an attempt to solve the above bottleneck problems and meet the demands of commercialization. Therefore, it is important to understand critical factors that influence γ-PGA microbial synthesis in depth. This review focuses on production strains, biosynthetic pathway, and metabolic regulation. Moreover, it systematically summarizes the functional properties, purification procedure, and industrial application of γ-PGA.

## 1. Introduction

γ-PGA is a biopolymer of D- and/or L-glutamic acid monomers by peptide bonds [1,2,3]. Commonly, γ-PGA is natural, water-soluble, edible, non-immunogenic, and biodegradable. Nowadays, due to specific biological properties, γ-PGA has already gained significant attention and has been widely applied into the fields of food processing, agricultural production, emerging medical treatment, cosmetics, and other fields [4,5,6,7,8,9,10,11,12,13]. 

Microbial synthesis is an extremely economical and effective way to produce γ-PGA [11]. Bacteria, archaea, and eukaryotes are all γ-PGA producers, but *Bacillus* species are the natural, dominant, and safely produced strains [7,8,14,15,16]. Commonly, the production strains are divided into glutamic acid-dependent and glutamic acid-independent strains according to whether there is a need to add glutamate acids externally [17]. Glutamic acid dependent strains are regarded as the better producers, as adding glutamic acids can significantly increase the yield [1,9,18]. Biosynthesis of γ-PGA is well established, but both productivity and yield are still the decisive factors for limiting its industrial application [11].

The molecular composition, molecular weight, yield, and synthesis productivity of γ-PGA directly affect its biological properties and industrial applications. Most recent research have been paid attention to metabolism regulation of γ-PGA synthesis. Some of the strategies may be promising in the future, including genetic manipulation, culture medium, and culture condition optimization, which are all expected to improve the yield and productivity [11]. 

This review mainly summarizes the recent advances about the molecular components, functional properties, microbial sources, synthetic pathway, metabolic regulation, purification processes, and industrial application of γ-PGA. Meanwhile, it reveals tough challenges of γ-PGA biosynthesis ahead and strategies for challenges.

## 2. The Functional Properties of γ-PGA

γ-PGA is polymerized from D-/L-glutamic acid monomers alone or both enantiomers by peptide bonds, which are synthesized by the α-amino of one glutamic acid and the γ-carboxylic acid of another one (Figure 1) [19,20,21]. The peptide bonds of γ-PGA are significantly different from the amide bonds of ordinary proteins, which are formed by one amino acid with α-amino group and another one with α-carboxylic acid group [19].

The molecular structure of γ-PGA is directly affected by the molecular composition, molecular weight, and order of the isomers. Based on the monomer types, γ-PGA consists of γ-L-PGA (homopolymers containing only L-glutamic acid), γ-D-PGA (homopolymers containing only D-glutamic acid), and γ-DL-PGA (random copolymer containing D- and L-glutamic acid) [8,12]. Generally, γ-PGA covers five types of structures: α-helix, β-sheet, random coil, helix-to-random coil transition, and enveloped aggregate [12,22,23]. The molecular structures are always influenced by the extraction process [1]. Degree of monomer polymerization reflects the molecular weight of γ-PGA, which is determined by the producers, media, and cultivation conditions [24,25]. The molecular weight of γ-PGA is generally in the range of 10 kDa to 2000 kDa [19]. The higher molecular weight, the higher viscosity and the more difficult it is for further purification preparation [3,5,11,12,26,27].

The structure and molecular component of γ-PGA decisively influence its functional properties. As an anionic biopolymer with both carbonyl and amide groups, γ-PGA possesses an array of specific properties: edible, strong water solubility and water retention, non-toxicity, high biodegradability, high cation exchange capacity, metal-chelating ability, strong antioxidant, antimicrobial peptides activities, and high resistance to thermal decomposition [21,28]. However, some functional properties of γ-PGA are not completely understood. Depending on the biological properties, γ-PGA might be used as food additives, e.g., taste-masking agents, texture modifier, thickeners, stabilizers, moisturizers, and probiotics cryoprotectant [12,22,29]. It may be an essential material for maintaining sensitive activities and is used as carriers, fillers, and adhesives in complex matrices, such as for drugs in the medical industry and tissue engineering [30,31,32,33,34,35,36,37,38]. Besides, in agriculture, it can be an ideal stabilizer, metal biosorbent in soil washing, an environment-friendly fertilizer synergist, and plant growth accelerator [24,38,39,40]. Additionally, it can be used in moisturizers in the cosmetics industry [41].

## 3. Microbial Sources of γ-PGA

Since γ-PGA was first discovered in a capsule of *Bacillus anthracis* by Bruckner and his co-workers in 1937, it has been discovered successively in archaea, bacteria, and eukaryotes, such as *Natrialba aegyptiaca*, *Natronococcus occultus*, and *Fusobacterium nucleatum* [14,15,42]. However, natural sources of γ-PGA have been occurring in the fermented products of *Bacillus* species for a long time, which contains *B. subtilis*, *B. amyloliquefaciens*, *B. licheniformis*, *B. velezensis*, and others [16]. *B. subtilis* and *B. licheniformis* are high potential microbial sources of γ-PGA [3,11]. *Bacillus* species, such as γ-PGA producers, can be separated into two types: glutamic acid-dependent and glutamic acid-independent strains; the former need an external supply of glutamate to produce γ-PGA, and the latter can produce γ-PGA without extra adding glutamate [1,9]. Most γ-PGA producers are known as glutamic acid-dependent strains, such as *B. subtilis* F-2-01, *B. subtilis* MR-141, *B. subtilis* C10, *B. subtilis* chungkookjang, *B. subtilis* GXA-28, *B. licheniformis* ATCC 9945a, *B. licheniformis* A35, and *B. licheniformis* WX-02 [24,43,44,45,46,47,48]. Relatively few strains, such as *B. subtilis* TAM-4, *B. subtilis* C1, *B. subtilis* 5E, *B. licheniformis* A35, *B. licheniformis* S173, and *B. amyloliquefaciens* LL3, are glutamic acid-independent strains [11,42,49,50,51]. Although, compared to glutamic acid-independent strains, glutamic acid-dependent strains are generally better producers and have attracted much attention to relatively high yield of γ-PGA, the current problem is high cost of production [1,9,10]. Instead, glutamic acid-independent strains are more desirable for realistic demands of yield and productivity. They mean not only low cost but also simple fermentation processes. However, lower γ-PGA productivity is the primary limitation [3]. 

Nowadays, most commercial γ-PGA is mainly synthesized by *Bacillus* species and safely used in various well-established processing fields [1,4,5,8,10]. *B. subtilis*, as a traditional model organism, is a class of gram-positive, endospore-forming, and rod-shaped bacteria. It has applications in such fields as food and medicine as a Generally Recognized as Safe (GRAS) microbial producer. Furthermore, most importantly, it possesses the best-characterized γ-PGA production abilities [1,11]. Both the genome and molecular genetics relationship of *B. subtilis* have been clarified, and the manipulation is well established. *B. licheniformis* is another type of gram-positive, endospore-forming bacteria. At present, it has been exploited for γ-PGA production [1]. In addition, *B. methylotrophicus*, *B. anthracis*, and *B. thuringiensis* also have the great productivity of γ-PGA [26,52]. However, these strains have less relevant studies about γ-PGA production [53]. They all cannot be the great producers because of some natural flaws; e.g., *B. anthracis* is a pathogenic bacterium, and it is not viable for industrial applications.

In fact, engineered microorganisms have also been used as a suitable device for γ-PGA biosynthesis [1,11,54,55]. *Escherichia coli* are the most used host for heterologous expression. *C. glutamicum* is a good native L-glutamic acid producer [5,19]. Therefore, it is regarded as a host for producing recombinant γ-PGA. Similarly, *B. subtilis* can also be the homologous host engineered to produce γ-PGA [5,19]. However, the final yield of γ-PGA is and remains far below that of native strains.

## 4. Synthetic Pathway of γ-PGA

The microbial synthetic pathway of γ-PGA principally covers three distinct stages: racemization, polymerization and transfer, and catabolism (Figure 2) [10,11,22]. Glycolysis, the Pentose Phosphate Pathway (PPP), Tricarboxylic Acid (TCA) cycle, amino acid metabolism, and glutamate synthesis all participate in γ-PGA biosynthesis [56].

### 4.1. The Racemization Pathway of γ-PGA

Racemization is the first step of anabolism. Under the catalytic action of glutamate racemase (RacE), exogenous or endogenous L-glutamate are converted directly to D-glutamate [11]. Then these glutamic acids monomers incorporate into the growing L-chain [15,21]. In *B. subtilis*, *racE/glr* and *yrpC* have been identified as two homologs of RacE coding genes, and *racE/glr* has proven to be essential for the direct conversion between L-glutamate and D-glutamate [57]. For indirect conversion pathway, with pyruvate as the precursor, three enzymes catalyze the transformation of pyruvate to D-glutamic acid (L-glutamic acid/pyruvic acid aminotransferase, D-glutamic acid/pyruvic acid aminotransferase, and alanine racemase) [1,20]. 

In the presence of large amounts of L-glutamic acid, *Bacillus* species will initiate the transformation pathway to produce D-glutamic acid. In L-glutamic acid-dependent strains, once the L-glutamate is imported from the medium into the cytoplasm, part of them is immediately converted to D-glutamic acid and involved in the downstream synthesis pathway [3]. In L-glutamic acid-independent strains, all L-glutamic acid required is generated from external carbon sources [19]. With catalytic activity of glutamate dehydrogenase (GDH), these strains produce L-glutamic acid by transforming citric acid into both isocitric acid and α-ketoglutaric acid in TCA cycle [19]. Or L-glutamic acid production relies on glutamine synthetase-glutamate synthase (GS-GOGAT) pathway. Pyruvic acid and α-ketoglutaric acid are mutually transmitted by aminotransferase [58]. *C. glutamicum* has also been well-established for γ-PGA biosynthesis. Clearly, it has two completely different pathways: the GS-GOGAT pathway regulated by GlnA, GltB, and GltA and the NADPH-dependent pathway regulated by RocR, RocG, and GudB (RocR, a transcriptional regulator of glutamate; RocG, expressed by *rocR*; GudB, glutamate dehydrogenase) [3,19].

### 4.2. The Polymerization and Regulation Pathway of γ-PGA

The next steps are polymerization and transfer. L-glutamic acid and D-glutamic acid monomers are transfered from cytoplasm to cell membrane with the membrane-embedded γ-PGA synthetase (Pgs) [10,11,18]. Pgs are responsible for assembling them into γ-PGA [5]. Pgs is encoded by an active membrane enzyme complex (four gene subunits operons: *pgsB*, *C*, *A*, and *E*) and the γ-PGA-release gene (*PgsS*) (Figure 3) [2,59]. PgsBCA has been identified as the unique mechanism of γ-PGA microbial synthesis in *Bacillus* species [5,60]. The expression level of PgsBCA has a direct effect on γ-PGA biosynthesis [19]. PgsB and PgsC make up the main components of the catalytic site. PgsA removes the long chains on the active site; it participates in the integration of γ-PGA long chains and later transfer [13,60]. PgsBCA is necessary for polymerizing and transporting γ-PGA to the compact cell membrane [12]. High concentrations of PgsB, PgsC, and PgsA can biosynthesize γ-PGA from a lack of PgsE; therefore, PgsE may be dispensable [60]. However, PgsE is essential for γ-PGA biosynthesis when Zn^2+^ is present because PgsBCA complex is extremely unstable and highly hydrophobic, and PgsE could affect PgsBCA disaggregation in *Bacillus* species [61]. The homologs of PgsBCA are YwsC and YwtAB in *B. subtilis* and CapBCA in *B. anthracis* [1,45]. γ-PGA can be anchored at the peptidoglycan of cellular surface with CapD catalysis or released into the medium with PgsS catalysis [52]. 

Meanwhile, γ-PGA biosynthesis is regulated by two intracellular signal transduction mechanisms: one is ComP-ComA regulator, and another is DegS-DegU regulator [62]. DegQ regulated by these two mechanisms inhibits the γ-PGA production and then effectively down-regulates the expression level of degradation enzymes [63]. SwrA is another regulator of PgsBCA. The fully activated PgsBCA operon needs the simulaneous presence of SwrA and phosphorylated DegU (DegU-P), but one of the two exists separately and does not have much impact on *pgsBCA* transcription [18,64]. High level of DegU-P could completely replace SwrA in directly activating PgsBCA expression [65]. Overall, these mechanisms are all involved in transcriptional regulation. Both ComP-ComA and DegS-DegU systems regulate the front-end transcription, while SwrA assists in regulation at the post-transcription [66].

### 4.3. The Catabolism Pathway of γ-PGA

γ-PGA is a secondary metabolite for the *Bacillus* species. However, production strains of γ-PGA may possibly degrade and utilize γ-PGA as substrates [24]. The catabolism pathway of γ-PGA is rigorously controlled by the intracellular regulatory mechanisms [19].

Three types of hydrolases can degrade γ-PGA: (1) native γ-PGA hydrolase (PgdS), whose coding gene is located in the downstream of PgsBCA operon, cleaves the γ-glutamyl bonds with strict regulation inside of hosts; (2) cell wall lyases (D- or L-endopeptidase), such as CwlO, CwlS, LytE, and LytF, disconnects between the glutamic acid residues peptide bonds; and (3) γ-glutamyl transferase (excision or incision enzyme, Ggt) hydrolyzes the γ-PGA from the N-terminal and releases shorter D- or L-glutamic acid residues to form di- and tripeptides of γ-glutamic acid in vitro [56,67,68,69].

The genes encoding Ggt (*PgdS*, *YwtD*, and *Dep*) are in plasmid and lie directly in the downstream of PgsBCA operon in the *Bacillus* species with the same orientation [67]. The above situation indicates that the enzymes are stably expressed under the ComP-ComA and DegS-DegU regulators.

## 5. Improvement of Microbial γ-PGA Synthesis

Although microbial synthesis of γ-PGA has been well developed, yield and productivity remain the core constraints for its industrial application [11]. Most cost of γ-PGA production depends on culture medium and producing strains [70]. Metabolic regulation, including genetic manipulation, culture medium, and culture condition optimization, is an effective way to solve the above bottleneck problems. However, the final yield of γ-PGA has so far not matched and even falls far below the productivity of the native producers [71].

### 5.1. Genetic Manipulation

Thus far, most native producers of γ-PGA are *Bacillus* species. Parts of them have great productivity. *B. subtilis* has been widely applied in industry by simple screening and improvement (Table 1 and Table 2) [1]. Production strains of γ-PGA have always been isolated from traditional fermented soybean products or some specific environments. However, the high production cost of glutamate-dependent strains and low substrate conversion rate of glutamate-independent strains should not be ignored. Genetic manipulation is a proven effective way to regulate the γ-PGA production [19]. Genetic manipulation is the key process of constructing genetically engineered strains. It mainly contains knock-out genes and construction plasmid. Genetic manipulation can promote or inhibit expression of enzymes and thus influences γ-PGA biosynthesis and degradation directly or indirectly in *Bacillus* species.

For native strains, genetic manipulation targeting γ-PGA synthesis pathway is a common strategy. Deletion the transcriptional regulating gene *rocR* or the glutamate dehydrogenase gene *gudB* individually can effectively increase the yield of γ-PGA in *B. amyloliquefaciens* LL3 [83]. In *B. amyloliquefaciens*, NK-A6 deleted *fadR*, *lysC*, *aspB*, *pckA*, *proAB*, *rocG*, and *gudB* for partial blockade of downstream metabolic pathways; *B. amyloliquefaciens* NK-A7 inserted a strong promoter P_C2up_ for enhancing NADPH level by genetic manipulation; and *B. amyloliquefaciens* NK-A11 deleted the srf and itu operons, and these engineered bacteria synthesized 4.84, 6.46, and 7.53 g/L γ-PGA, respectively, all higher than the original strains of *B. amyloliquefaciens* LL3 [84]. Some strains, such as *B. subtilis* 168 and *B. subtilis* DB430, have the necessary synthesis gene cluster but cannot produce γ-PGA [59,85]. A mutant integrated PgsBCA or ywsC-ywtAB gene cluster with the regulating gene *SwrA* can increase intracellular level of DegU-P and SwrA and the production of γ-PGA. Besides, PgsBCA embedded into chromosome with a strong promoter can efficiently enhanced the production of γ-PGA. 

Moreover, γ-PGA may be hydrolyzed and acts as a carbon or nitrogen source by production strains of γ-PGA if lacking external nutrient supply [19]. Repressing γ-PGA hydrolase by gene engineering could reduce loss of production and change molecular weight of γ-PGA [39,78]. Some research paid attention to the expression of γ-PGA hydrolases manipulated by gene knockout. Compared to wild strains, mutant deleted *ggt* in *B. subtilis* NAFM5 had higher molecular weight, but the yield of γ-PGA might be comparable [86]. In *B. subtilis* and *B. amyloliquefaciens*, deleting *cwlO* gene increased production even with higher molecular weight of γ-PGA relative to the wildtype strain [77], and deleting *pgdS* gene just increased molecular weight slightly [69]. However, the combined deletion of *pgdS*, *cwlO*, or *ggt* in the above strains resulted in a 93% increase in production compared to the wildtype [65,80]. Besides, deleting *luxS* gene in *B. amyloliquefaciens* NK-E10 resulted in a slightly improved production ability of γ-PGA [82]. 

In addition, enhancement of carbon flux conversion to γ-PGA biosynthesis, elimination of by-products or undesired precursor, and inhibition of other drain energy pathways can promote γ-PGA production and purity [42]. Carbon metabolism may be impacted positively by extra ATP, then improved by γ-PGA production [87]. Polysaccharides, major by-products and contaminates of γ-PGA, consume large amounts of the carbon sources and metabolic energy [19]. Mutants knocked out the synthetic genes of exopolysaccharides and lipopolysaccharide to obtain more and purer γ-PGA [82,88]. Compared with no removal of lipopolysaccharides in medium, the purity of γ-PGA in *B. amyloliquefaciens* NK-E5 increased to 95% [82]. Similarly, mutants removing the synthetic genes of both lactate and acetate, toxic compounds adversely impact cell growth, produced the same effects as the above study. However, approaches of eliminating glutamate-consuming pathways did not increase the yield of γ-PGA because the gene manipulations resulted in a metabolic imbalance with other related pathways in cells [19,82]. Just when γ-PGA is biosynthesized, energy-consuming substances synthesis also take places in cells, such as antibiotics. Repression of these energy-consuming substances synthetic pathways by gene knockout may increase production relative to the native strains but may not always be helpful [89].

Recombinant expression in homologous or heterologous hosts is another effective strategy for increasing γ-PGA production (Table 3). Genetic recombination is divided up into homologous expression and heterologous expression. Genetic manipulation recombined by *E. coli* and *C. glutamicum* is heterologous expression. Genetic manipulation recombined by *Bacillus* species is homologous expression. The xylose-induced plasmid pWH1520 with *pgsBCA* operon can be introduced into *B. subtilis* MA41 with the disruption of native *pgsBCA* gene, and then, γ-PGA synthetase can be successfully expressed [54]. Plasmid harboring the energy-saving NADPH-GDH pathway in *B. amyloliquefaciens* NK-1 could increase the yield of γ-PGA by 9%. Genetic expression patterns, including constitutive and inducible expressions, are the key factors affecting γ-PGA yield [78,90]. Inducible expression can accumulate γ-PGA in a short time. Besides, the operon *pgsBCA* of *B. licheniformis* NK-03 and *racE* of *B. amyloliquefaciens* LL3 were simultaneously cloned into an induced plasmid and co-expressed in *E. coli* JM109 for microbial synthesis of γ-PGA. Constitutive promoter (PHCE) from the D-amino acid aminotransferase (D-AAT) gene of *Geobacillus toebii* was recombined and expressed efficiently in *E. coli*/pCOpgs for γ-PGA biosynthesis. The final yield of γ-PGA is 3.7 g/L in the optimized medium [91].

Among others, *C. glutamicum* may be considered as a suitable host for recombinant production of γ-PGA, which is a native L-glutamate producer. When glutamate was deficient, heterologous expression of PgsBCA operon still resulted in the yield up to 0.7 g/L for *C. glutamicum* [19].

### 5.2. Culture Medium Optimization

Optimized medium composition can effectively promote cell growth and thus accumulate large amounts of precursors for γ-PGA synthesis (Table 4) [71]. More importantly, engineered strains that metabolize cheap substrates are an effective strategy for cost reduction of γ-PGA production. Carbon sources as major substrates result in directly or indirectly influence in γ-PGA production. Glucose, sucrose, glutamic acid, citric acid, and glycerol are the common carbon sources with different metabolic pathways and influences for γ-PGA production [95]. Glucose and citric acid are the most utilized carbon sources by *Bacillus* species [96]. Glucose metabolism is related to protein metabolism, such as several stress-response proteins [97]. Carbon catabolite may affect γ-PGA synthetic enzyme activity. Glucose may suppress the transcription of *degQ* operon during γ-PGA synthesis. Both citric acid and glutamic acid could serve as promoters for γ-PGA production [98]. Though fructose, maltose, and xylose show a positive effect on cell growth, almost no change was shown in γ-PGA production. Adding metabolic precursors of γ-PGA production as carbon sources may obtain higher yields, possibly because of enzymes activity enhanced of metabolic pathways [42]. Culture medium with L-glutamine and α-ketoglutaric acid can effectively increase the yield of γ-PGA up to 20% for *B. subtilis* BL53 [99]. The molecular weight also increased to 570 kDa [100]. Citric acid reached maximum yield to 28.3 g/L when compared with other organic acids, such as succinic acid, malic acid, and oxalic acid [47]. Organic acids involved in metabolic pathways often impact γ-PGA biosynthesis by adjusting enzymes activities. Most glutamic acid-independent strains prefer glucose and glycerol. Glycerol never inhibits related enzymes for γ-PGA production [97]. Compared with glucose, cells in glycerol can be more permeable, which may help the produce and release of γ-PGA [101,102]. Other biomass materials or by-products also can efficiently convert into high-value γ-PGA. Agro-industrial wastes were exploited as raw materials, such as cane molasses, corncobs, rapeseed meal, soybean residue, monosodium glutamate for processing wastewater, rice straw, and crude glycerol and its hydrolysate [74,96,103]. Besides, some carbonaceous substances might replace common carbon sources, such as algae, chicken manure, animal feathers, and dairy products [104,105,106,107]. The yield of γ-PGA reached 65 g/L in an optimized culture medium containing glutamic acid extracts, soybean residue, and chicken manure [108].

The nitrogen-containing compounds also can affect γ-PGA production. NH_4_^+^, an inorganic nitrogen source, easily form monomers of γ-PGA with α-ketoglutarate by transamination reaction [118]. The synthetic pathway of γ-PGA needs free amino groups. Glutamate is the most common nitrogen source for *B. subtilis*. Most L-glutamic acid is metabolized as a nitrogen source, and only a small amount is used to synthesize γ-PGA [119]. In addition, others nitrogenous substances are also suitable alternatives of nitrogen sources. With fishmeal waste as the nitrogen source in optimized culture medium, the yield of γ-PGA is increased up to 25 g/L [24].

Inorganic salts can also affect γ-PGA production. Metal ions serve as cofactors in utilization of carbon sources. Mn^2+^ can significantly enhance the glutamate racemase activity of γ-PGA, and then, the yield of γ-PGA can increase from 9.25 to 28.42 g/L [20,98]. The molecular weight and yield of γ-PGA can be regulated by sodium salts of culture medium for halotolerant strains as well as in other extremophilic bacteria [46]. Besides, γ-PGA derivatives with phosphorylation, esterification, or sulfonation can be directly produced by adding agents into the growth media, such as phosphoric acid. γ-PGA derivatives have greater potential than native γ-PGA and have been unprecedentedly used in various fields [7].

Exogenous additives of the culture medium are useful for γ-PGA production. Additives influence cellular membrane permeability and then regulate substrate availability. Tween 80 and DMSO can increase utilization of carbon sources [118].

It has been proven that suitable culture conditions, such as temperature, oxygen content, pH, and the inoculation amount. can effectively improve the yield of γ-PGA. Most *Bacillus* species are mesophilic and aerobic strains, tend to thrive well on optimum temperature range, and effect γ-PGA synthesis [48]. Some thermoacidophiles still maintain thermal stability in high temperature [120,121]. In addition, some strategies for maintaining oxygen supply, such as adding oxygen-carrying agent or one cultured in a fermenter, can enhance the γ-PGA production [99]. Another important factor is pH for γ-PGA biosynthesis because reactions involving glutamate are pH sensitive [11]. pH-controlling strategy further increases the utilization of glutamate and hence improves the γ-PGA production. The amount of inoculation is determined by the type of bacteria, culture composition, and culture conditions [122]. Inappropriate concentration of inoculum causes excessive nutrient competition or inhibition of the growth of viable cells and thus would reduce γ-PGA biosynthesis.

The optimization and integration of various conditions is the focus of current research, which take less fermentation time, has lower production costs, and has more stable productivity and yield. A series of methods, such as batch, fed-batch, continuous culture, and symbiotic fermentation (mixed strains or substrates are co-cultured), were tested at small scale to optimize the productivity of γ-PGA [8,54,123,124]. Clearly, these combined methods are highly effective. The yield of γ-PGA was increased to 2.19 g/L in a fermenter by fed-batch of glucose supply [72].

## 6. Purification of γ-PGA

γ-PGA is synthesized by *Bacillus* species outside the cells, which makes its purification relatively simple. The yield and molecular weight of γ-PGA, determined by biosynthesis, are the key factors affecting its purification [118]. Moreover, it has been confirmed that the purity of γ-PGA relates to culture mediums and conditions [112,125]. The higher the yield or/and the molecular weight of γ-PGA, the higher the viscosity and the more difficult the purification is [11]. Besides, part of γ-PGA is adhered to cell wall peptidoglycan rather than being secreted into the medium, which apparently makes purification of γ-PGA more difficult [1,14]. Precipitation and filtration are commonly used to purify γ-PGA. The first step is removal of the cells of fermented broth before purification, which can simplify the purification process. Diluting the medium or adjusting the pH value to about 3 can reduce the viscosity and help to remove the producing strains [126]. Then, the broth is removed by elementary precipitation and filtration. The normal method is organic solvents precipitation [42]. However, some of water-soluble organic solvents will degrade γ-PGA into oligomers or monomers. Ethanol remains the preferred precipitator of γ-PGA precipitation from a cell-free broth [126]. However, it results in the co-precipitation of other macromolecular substances, such as proteins and polysaccharides [11,26]. Obviously, further effective purification may be required for separation, such as filtration and buffer exchange, according to the physicochemical properties’ differences in γ-PGA and other constituents [116]. Diafiltration and ultrafiltration are commonly used and cost-effective filtration technologies [126]. Continuous fermentation integrated with a membrane-based filtration technology has been applied to industry [127]. However, membrane-based filtration technique is usually exceedingly slow. Besides, metal ions also can be used to precipitate and isolate γ-PGA, and Cu^2+^ has been proven to be an effective precipitator at 500-mM concentration with recovery of 95% [42]. Still, most of the metal ions are toxic, and it is also possible to contaminate the γ-PGA and environment [11,128]. Next, crude γ-PGA can be extracted by centrifugation and redissolved in distilled water for dialysis. The final step is lyophilization for obtaining purified γ-PGA [100].

## 7. Application of γ-PGA

Due to the various biological characteristics, such as water-solubility, thickness, moist quality, edible nature, non-toxicity, non-immunogenic nature, and antioxidant and antimicrobial activities., γ-PGA has been applied extensively in the food field, medical field, in environmental engineering field, and other industrial fields [4,5,6,7,8,9,10,11,12,13]. 

### 7.1. Food Industry

In the food industry, as an edible, non-toxic, strong water-retaining antioxidant and antimicrobial biopolymer, γ-PGA can replace the existing as a kind of emerging dietary supplement in functional food. As a texture enhancer and stabilizer, γ-PGA can improve the quality of starch, wheat gluten (WG), and their containing products [31]. For fermented protein products, γ-PGA can stabilize the viability of microorganisms, texture, aroma, and flavor of foods during the production, transportation, and sale of products, such as yoghurts [129]. It has been shown that γ-PGA can relieve bitter taste as a bitterness-masking agent in substances of amino acids, peptides, quinine, caffeine, minerals, etc., and as a thickener added to fruit juice beverages [82]. Based on the anti-freezing activity, γ-PGA is also a cryoprotectant to preserve the viability of probiotics during freeze-drying [130]. γ-PGA, with a molecular weight of lower than 20,000 Da, has higher anti-freezing activity than glucose and is more effectively protective to *Lactobacillus paracasei* than sucrose, trehalose, and sorbitol [29,81]. Furthermore, γ-PGA can increase the Ca^2+^ bioavailability and effectively prompt intestinal absorption [131]. Besides, the study of its antioxidant mechanism has demonstrated that γ-PGA can protect the gastrointestinal tract against oxidative damage [40].

### 7.2. Agricultural Field

γ-PGA shows great promise in agroecosystem due to its biological properties of water solubility, strong water retention, high biodegradability, high cation exchange capacity, and metal-chelating ability. For instance, γ-PGA is increasingly garnering interest in soil dynamics and plant growth as an environment-friendly fertilizer synergist [104]. γ-PGA may improve uptake of water-soluble nutrients and increase both root biomass and activity [28]. γ-PGA can be exogenously applied to protect seedlings from the adverse effects [132]. Moreover, based on biocontrol capacity and antimicrobial activities, γ-PGA has been used in abiotic or biotic stress resistances against hostile conditions [133]. As a biological chelating agent, γ-PGA is extremely beneficial to alleviate toxic accumulation caused by heavy metals of crops and farmland pollution [134].

### 7.3. Bio-Medical Field

γ-PGA has exhibited great potential in pharmaceutical manufacturing, which involves tissue engineering, delivery systems, and preventive, immunological, and therapeutic effects [11,135]. γ-PGA, a natural, non-toxic, and non-immunogenic biomacromolecule, can significantly decrease the toxic effects and improve the efficiency of drugs when combined with other matters [11]. Especially, biodegradable γ-PGA has a bright potential in drug-delivery platforms and as suitable carriers for gene therapy [81]. Molecular weight of γ-PGA was the decisive factor for considering the drug-delivery properties, including controlling or delaying the release [11]. γ-PGA has been used as new biological adhesives, formed by chemical cross-linking with other substances [40]. Besides, γ-PGA can induce a higher immunogenicity of vaccine antigens as a good adjuvant [136]. Furthermore, γ-PGA has showed excellent protection of Caco-2 cell and probiotics from oxidative damage [137].

### 7.4. Other Fields

γ-PGA is a natural, high-performance, hydrophilic humectant and dispersant in cosmetics because of its hydroscopicity, surface adhesion, antioxidant effects, and cytoprotection [41]. γ-PGA is expected to be the most useful flocculating agent due to its biodegradability and non-toxicity toward humans and the environment, which is superior to other conventional agents [19,138]. γ-PGA will be utilized not only in biomass waste and wastewater treatments in the industry but also in domestic water processing [81]. Besides, numerous chemically modified derivatives of γ-PGA have been developed so far as biopolymers utilized as novel biomaterials [12]; e.g., esterified γ-PGA can be used as an excellent thermoplastic for its capability to form biodegradable fibers and film [44,81].

## 8. Conclusions

γ-PGA is a promising biomacromolecule constituent of D- and/or L-glutamic acids, which has been applied worldwide in various industrial sectors, such as food processing, agricultural production, emerging medicine, environmental engineering, the cosmetic industry, and other related fields. Currently, microbial synthesis is a commonly used method to produce γ-PGA, and *Bacillus* species are natural, dominant, and safely produced. The molecular composition, molecular weight, synthesis productivity, and yield of γ-PGA directly affect its biological properties, which finally determined its industrial applications. At present, with the gradual increase in industrial demand, γ-PGA is of great interest for potential bio-based materials even if may impact existing industries and conventional commercialized polymers. Therefore, though microbial synthesis is the extremely economical and effective way to produce γ-PGA, cost of production and yield are the major constraints of industrial applications. The improvement of γ-PGA biosynthesis, including potential microbial producers screening, genetic manipulation, culture medium, and culture condition optimization of the producing strains, can increase yield and reduce production costs. The subsequent research will certainly be focused on novel theoretical studies and advanced technologies of γ-PGA microbial synthesis in the future.

## Figures and Tables

**Figure 1 foods-11-00739-f001:**
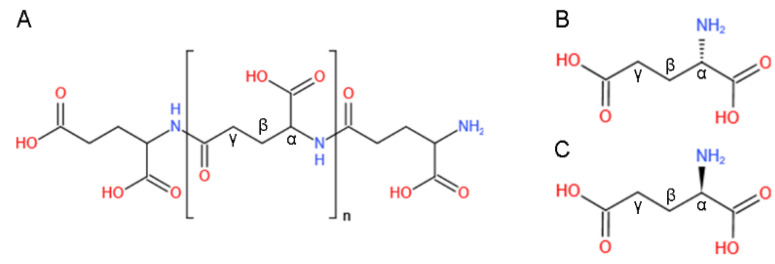
The structural formula of γ-PGA and its constituent units. The polymer of γ-PGA (n: repeating units approach at least 10,000) (**A**) and the L-glutamic acid monomer (**B**) and D-glutamic acid monomer (**C**) of γ-PGA.

**Figure 2 foods-11-00739-f002:**
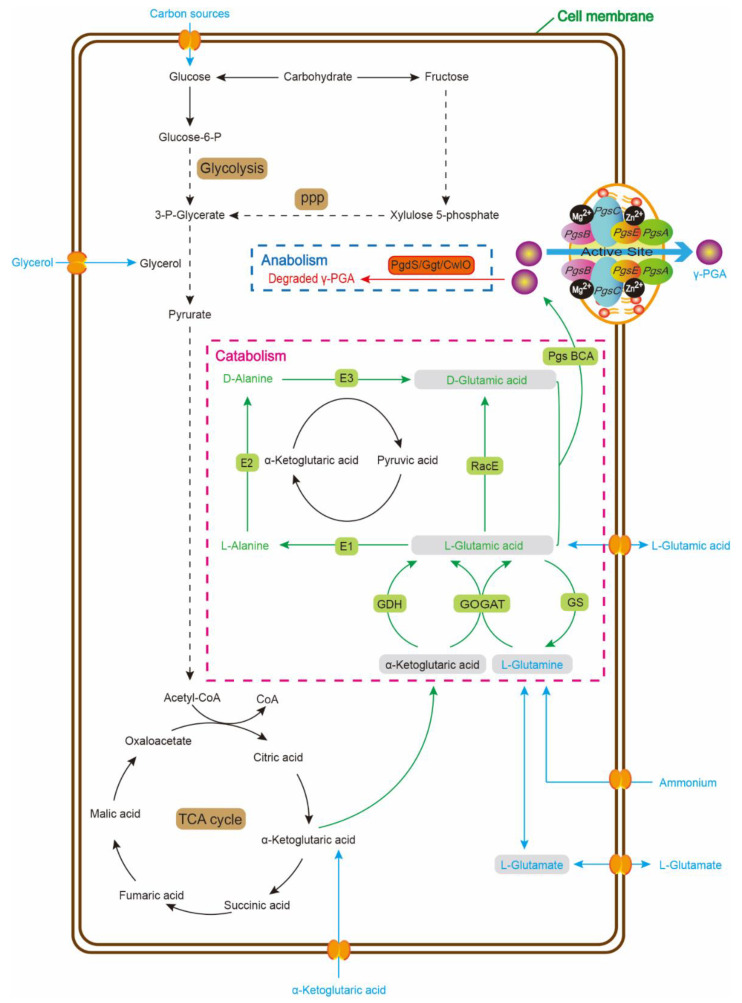
The synthetic pathway of γ-PGA. Main biologically synthetic pathway of γ-PGA involves three steps: racemization, polymerization and transfer, and catabolism. Various substrates (e.g., glucose, citric acid, glycerol, biomass materials, and by-products) enter by the main anabolism pathway. For glutamate-independent strains, they can utilize exogenous nutrients to product γ-PGA through glycolysis, PPP, and TCA cycle. Instead, for glutamate-dependent strains, glutamic acid or glutamate are added to the medium directly to product γ-PGA. PPP, Pentose Phosphate Pathway; G3P, glyceraldehyde 3-phosphate; TCA cycle, Tricarboxylic Acid cycle; E1, pyruvic acid aminotransferase (L-Glutamic acid); E2, alanine racemase; E3, pyruvic aminotransferase (D-Glutamic acid); RacE, glutamate racemase; Pgs, γ-PGA synthetase (four gene subunits operons: *pgsB*, *C*, *A* and *E*); GDH, glutamate dehydrogenase; GS, glutamine synthetase; GOGAT, glutamate synthase or glutamine oxoglutarate aminotransferase. PgdS, γ-PGA hydrolase; Ggt, γ-glutamyl transferase; CwlO, cell wall lyases.

**Figure 3 foods-11-00739-f003:**
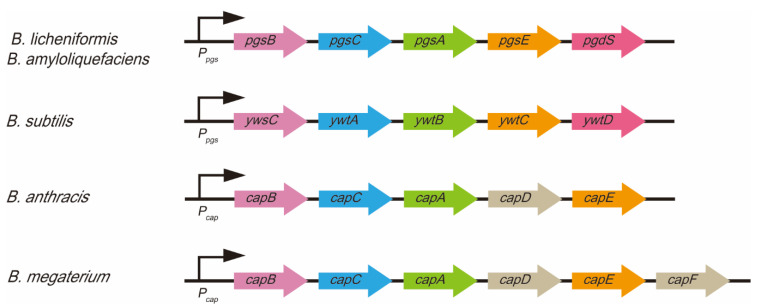
The synthetase gene operons for microbial γ-PGA in *Bacillus* bacteria. Pgs are responsible for assembling L-glutamic acid and D-glutamic acid units into γ-PGA. Pgs is encoded by an active membrane enzyme complex (*pgsB*, *C*, *A*, and *E*) and the γ-PGA-release gene (PgsS).

**Table 1 foods-11-00739-t001:** Sources of γ-PGA production trains.

Strains	Source	Main Medium Components	Cultural Conditions	Final Yield (g/L)	Ref.
*B. subtilis* ZJU-7	Isolated from fermented bean curd	Glucose, L-glutamic, yeast extract, NaCl, Ca^2+^, Mg^2+^, Mn^2+^	Bioreactor, pH 6.5, 37 °C	101.1	[72]
*B. subtilis* NX-2	Isolated from soil samples	Glucose, glutamic, (NH_4_)_2_SO_4_, yeast extract, K_2_HPO_4_, Mg^2+^, Mn^2+^	APFB (aerobic plant fibrous-bed bioreactor) immobilized cell fermentation, pH 7.0, 32 °C	71.21	[73]
	Isolated from soil samples	Cane molasses, monosodium glutamate liquid waste	Bioreactor, pH 7.0, 32 °C	52.1	[74]
*B. subtilis* MJ80	Isolated from soil samples	Glutamic acid, starch, urea, citric acid, glycerol, NaCl, K_2_HPO_4_, Mg^2+^, Mn^2+^	Fermenter for immobilized cell fermentation, pH 7.0, 37 °C	68.7	[75]
*B. subtilis* HB-1	Isolated from soil samples	Glutamate, xylose, corncob fibers hydrolysate, yeast extract, NaCl	Bioreactor, pH 6.5, 37 °C	28.15	[76]
*B. methyotrophicus* SK 19.001	Isolated from soil samples	Glucose, yeast extract, K_2_HPO_4_, Mg^2+^, Mn^2+^	Flask, pH 7.2, 37 °C	35.34	[27]
*B. licheniformis* P-104	Isolated from Chinese soybean paste	Glucose, glutamate, citric acid, (NH_4_)_2_SO_4_, K_2_HPO_4_, Mg^2+^, Mn^2+^	Bioreactor, pH 7.0, 37 °C	41.6	[77]

**Table 2 foods-11-00739-t002:** Genetic manipulations of γ-PGA.

Strains	Engineering Methods	Main Medium Components	Final Yield (g/L)	Ref.
*B. subtilis* ISW1214	Carrying the plasmid of γ-PGA synthetic system	Sucrose, xylose, NaCl, NaHPO_4_, KH_2_PO_4_, Mg^2+^	9.0	[54]
*B. subtilis* PB5249	Deletion of genes (*pgdS* and *ggt*)	Glucose, L-glutamic acid, citric acid, NH_4_Cl, K_2_HPO_4_, Mg^2+^, Mn^2^, Ca^2+^, Fe^2+^	40	[78]
*B. licheniformis* WX-02	Expression of *glr* gene for encoding glutamate recemase	Glucose, L-glutamic acid, citric acid, NH_4_Cl, NaCl, K_2_HPO_4_, Mg^2+^, Mn^2+^, Ca^2+^, Zn^2+^	14.38	[79]
	Enhanced expression of *pgdS* gene	Glucose, glutamate, citric acid, NH_4_Cl, K_2_HPO_4_, Mg^2+^, Mn^2^, Ca^2+^, Zn^2+^	20.16	[80]
	Substituted by the native *glpFK* promoter with the constitutive promoter (*P43*), *ytzE* promoter (*PytzE*), and *bacABC* operon promoter (*PbacA*)	Sodium glutamate, citric acid, glycerol,	17.65	[81]
	Over-expression of *glpK*, *glpX*, *zwf*, and *tkt1* promoters	Sodium citrate, glycerol, NaNO_3_, NH_4_Cl	12.83	[81]
*B. amyloliquefaciens*	Deletion of gene (*epsA-O*, *sac*, *lps*, *pta*, *pgdS*, *cwlO*, *luxS*, and *rocG*), expression of synthetic small synthetic regulator RNAs (repressed the expression of *rocG* and *glnA* gene)	Sucrose, (NH4)_2_SO_4_, K_2_HPO_4_, KH_2_PO_4_, Mg^2+^	20.3	[82]
*B. amyloliquefaciens* LL3	Double knockout of gene (*pgdS* and *cwlO*)	Sucrose, (NH4)_2_SO_4_, NaCl, K_2_HPO_4_, KH_2_PO_4_	7.12	[68]
	Gene knockout of *rocR*, *rocG*, *gudB*, and *odhA*		5.68	[83]
	Gene knockout of *fadR*, *lysC*, *aspB*, *pckA*, *proAB*, *rocG*, and *gudB*	Tryptone, xylose, yeast extract, NaCl, ampicillin, chloramphenicol, or tetracycline	4.84	[84]
	Enhancing NADPH level by inserting a strong promoter *P_C2up_*		6.46	[84]

**Table 3 foods-11-00739-t003:** Genetic manipulations of γ-PGA.

Strains	Genetic Engineering	Main Medium Components	Final Yield (g/L)	Ref.
*C. glutamicum* ATCC 13869	Cloning and expressing *pgsBCA* of *B. licheniformis* TKPG011	Glucose, (NH_4_)_2_SO_4_, soy protein hydrolysate, thiamine hydrochloride, KH_2_PO_4_, Mg^2+^, Mn^2^, Fe^2+^, Ca^2+^	18	[92]
*C. glutamicum* ATCC 13032	Cloning and expressing *pgsABC* from *B. licheniformis* NK-03	Glucose, tryptone, yeast extract	0.7	[93]
*E. col**i* BL21	Cloning and overexpressing γ-PGA biosynthesis genes	Glucose, L-glutamic acid, yeast extract, (NH_4_)_2_SO_4_	3.7	[91]
*E. coli* LRP	Expressing *pgsBCA* and *race* from *B. amyloliquefaciens* LL3	Glucose, yeast extract, NaCl,	0.7	[1]
*E. coli* JM 109	Cloning *pgsBCA* and *racE* from both *B. licheniformis* NK-03 and *B. amyloliquefaciens* LL3 and co-expression	Glucose, L-glutamic acid, yeast extract, NaCl, (NH_4_)_2_SO_4_, K_2_HPO_4_, KH_2_PO_4_, Mg^2+^,	0.65	[55]
*B. subtilis* PB5249	*∆**pgdS**∆**ggt* deletion mutants	glucose, L-glutamate, citric acid, NH_4_Cl, K_2_HPO_4_, MgSO_4_·7H_2_O, MnSO_4_·H_2_O, FeCl_3_·6H_2_O, CaCl_2_·2H_2_O	40	[78]
*B. subtilis* WB600	Overexpressing *pWB980-pgsBCA*	Glucose, sodium glutamate, MgSO_4_, (NH_4_)_2_SO_4_, K_2_HPO_4_	1.74	[94]
*B. subtilis* ISW1214	Overexpressing *pWH1520-PxylA-pgsBCA*	Sucrose, xylose, NaCl, MgSO_4_, KH_2_PO_4_, NaHPO_4_	9.0	[54]
*B. licheniformis* WX-02	Overexpressing *pHY300PLK-P43-glr*	Sucrose, (NH4)_2_SO_4_, MgSO_4_, KH_2_PO_4_, K_2_HPO_4_	14.38	[79]
	Overexpressing *pHY300PLK-PpgdS-pgdS*	Glucose, sodium glutamate, sodium citrate, NH_4_Cl, MgSO_4_, ZnSO_4_, MnSO_4_, CaCl_2_, K_2_HPO_4_	20.16	[80]
*B. amyloliquefaciens* LL3	*∆**pgdS**∆**cwlO* deletion mutants	Sucrose, (NH_4_)2SO_4_, MgSO_4_, KH_2_PO_4_, K_2_HPO_4_	7.12	[68]
	*∆**rocR**∆**rocG**∆**gudB**∆**odhA* deletion mutants	Sucrose, (NH_4_)_2_SO_4_, MgSO_4_, KH_2_PO_4_, K_2_HPO_4_	5.68	[83]
*B. amyloliquefaciens*	*∆**cwlO**∆**epsA-Ovgb* deletion mutants	Sucrose, (NH4)_2_SO_4_, MgSO_4_, FeSO_4_·4H_2_O, CaCl_2_·2H_2_O, MnSO_4_·4H_2_O, ZnCl_2_, KH_2_PO_4_, K_2_HPO_4_,	5.12	[68]
	Repressed both *rocG* and *glnA* genes	Sucrose, (NH_4_)_2_SO_4_, MgSO_4_, KH_2_PO_4_, K_2_HPO_4_	20.3	[82]

**Table 4 foods-11-00739-t004:** The culture methods of γ-PGA.

Strains	Cultural Methods	Final Yield (g/L)	Ref.
*B. subtilis* NX-2	Using a two-stage strategy for agitation speed control	40.5	[109]
	Adding hydrogen peroxide	33.9	[110]
*B. subtilis* CGMCC 0833	Applying pH-shift control strategy	27.7	[111]
*B. subtilis* F-2-01	Adding more carbon sources (L-glutamic acid and glycerol)	45.5	[43]
*B. subtilis* BL53	Adding some precursors	25.2	[112]
*B. subtilis* C10	Abundant supply of organic acid	27.7	[47]
*B. subtilis* GXA-28	Addition of KCl	25.62	[113]
*B. licheniformis* TISTR 1010	Different feeding strategies (glucose, citric acid, NH_4_Cl, NaCl, Mg^2+^, Mn^2^, Ca^2+^, K_2_HPO_4_, Tween-80)	27.5	[114]
*B. licheniformis* ATCC 9945A	Different feeding strategies (L-glutamic acid, citric acid, glycerol, NH_4_Cl)	23.0	[98]
		35.0	[115]
		57.5	[116]
*B. licheniformis* WBL-3	Optimized culture medium (L-glutamic acid, citric acid, glycerol, NH_4_Cl)	22.8	[102]
	Optimized culture medium (glutamic acid, citric acid, glycerol)	19.3	[102]
*B. licheniformis* P-104	Optimized culture medium (glutamate, glucose, citric acid, (NH4)_2_SO_4_)	41.6	[77]
*B. licheniformis* NCIM 2324	Addition of metabolic precursors (α-ketoglutaric acid)	35.75	[18]
	Optimized culture medium (L-glutamic acid, citric acid, glycerol, (NH_4_)_2_SO_4_, K_2_HPO_4_, Mg^2+^, Mn^2+^)	26.12	[18]
	Optimization effecting factors at a time (L-glutamic acid, sugarcane juice, citric acid, NH_4_Cl)	35.75	[18]
*B. licheniformis* WX-02	Adding pH stress treatment	36.26	[101]
*B. velezensis* NRRL B-23189	Optimized culture medium (molasses, citric acid, NH_4_Cl)	4.82	[117]
*B. methylotrophicus* SK 19.001	Optimized culture medium (glucose, citric acid, NH_4_Cl)	35.3	[27]

## Data Availability

Not applicable.

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
