# Peer review of "Recent Advances in Microbial Synthesis of Poly-γ-Glutamic Acid: A Review"

_foods, 2022, doi:10.3390/foods11050739_

Round 1
Reviewer 1 Report
I have read the paper and I have identified some recommendations to the authors:
Did the authors consider the initial inoculum value (cfu / g or ml) when comparing the final PGA yield?
In section 2, 4, 4.2 source www.google.com in my opinion is insufficient.
Please, always use italics for the name of the gene.
In figure 2, please explain all abbreviations in the legend.
Author Response
Manuscript ID: Foods-1590602
Title: Recent Advances in Microbial Synthesis of Poly-γ-glutamic acid: A Review.
Authors: Danfeng Li, Lizhen Hou, Yaxin Gao, Zhiliang Tian, Bei Fan, Fengzhong Wang *, Shuying Li*
Foods
Dear Reviewers:
I only represent all co-authors, we are grateful to the editor and reviewers very much for their constructive comments on our manuscript (foods-1590602), and greatly thank you for considering the revised version of the manuscript. We have carefully considered all the reviewer’s comments and revised the manuscript accordingly. The revised sections are marked in red in the paper (Marked Manuscript: foods-1590602-v2). Detailed point-by-point responses are enclosed.
Attached please find the revised version, which we would like to submit for your kind consideration. And this is a response to your comments.
The major revisions are as follows:
- We have carefully revised the abstract and conclusion of manuscript.
- We have corrected errors in our manuscript.
- We have also revised and added to the highlights of manuscript more accurately according to reviewers’ comments.
- The list of references has been updated. We have referred to works published in recent years and adapted the relevant content of the manuscript according to the references.
We hope that the revised version is now suitable for publication in Foods: Probiotics, Prebiotics, Synbiotics, Postbiotics and Paraprobiotics – New Perspective for Functional Foods and Nutraceuticals, and we thank you for taking the time on this manuscript. The revised version has been approved by all co-authors of this paper.
Sincerely,
ShuYing Li, Associate Professor
Institute of food science and technology
Chinese Academy of Agricultural Sciences
Point-by-point responses to reviewers
Manuscript ID: Foods-1590602
Title: Recent Advances in Microbial Synthesis of Poly-γ-glutamic acid: A Review.
Reviewer #1:
Comment: I have read the paper and l have identified some recommendations to the authors:
Response: We are grateful for the positive comments from this reviewer. Our responses to the specific comments are set out below.
Comment 1: Q1: Did the authors consider the initial inoculum value (cfu / g or ml) when comparing the final PGA yield?
Author Response: Thanks for the Reviewer’s very kind suggestions. Though we do not reflect the initial inoculum value (cfu / g or ml) in the tables, we have considered the differences in inoculum value when comparing the final PGA yield. We compared the final PGA yield by production strains, culture medium, and culture methods, expect the initial inoculum value. Types of production strains, culture medium, and culture methods are more impact on final PGA yield. It is not optimized for the initial inoculum value in some related papers, and more attention to optimization of the culture medium. In addition, other reviews also did not reflect the initial inoculum value in the tables.
Comment 2. Q2: In section 2, 4, 4.2 source www.google.com in my opinion is insufficient.
Author Response: We are very sorry for the confusion to you. We have misquoted the URL in the paper. We have updated the relevant citations.
Comment 3. Q3: Please, always use italics for the name of the gene.
Author Response: Thanks for the Reviewer’s nice suggestion. We have corrected errors in our manuscript, such as line 247.
Comment 4. Q4: In figure 2, please explain all abbreviations in the legend.
Author Response: Thanks for the Reviewer’s nice suggestion. I have added the meaning of abbreviations in the legend.
Thank you very much for your comments and suggestions on our manuscript. We have tried our best to revise our manuscript in the light of the comments. Please find attached the revised version, which we would like to submit for your kind consideration.
Reviewer 2 Report
This review focuses on the recent development in the aspect of gamma-PGA production by microorganism. The topic is more related to the fields of biotechnology and biochemistry, rather than food field; hence, the suitability of this review paper for publication in Food journal is questionable. In specific, I do not find a good relevance of this topic with the food element, except for a small section about gamma-PGA application in food industry. The authors claimed to provide a review on the ‘recent’ advances, but most of the cited articles are prior to 2015. Please ensure that there is a sufficient development in this field before making a ‘recent’ review about the topic. In fact, the content covered by this review can be easily found in other available review papers (e.g., 10.1186/s13068-016-0537-7). In general, the English grammar is acceptable but there are a number of awkward sentences and typos to be corrected. Other specific comments are given below:
- Correct grammars in phrases such as: “It always been”, “can efficiently enhanced’, “can efficiently convert into”, “have be tested”, “Ca2+”.
- The citations for Figures 1-3 shows a general google website link. This is not the right way of acknowledging the source of citation. Do provide a proper citation in figure caption, and not in the main text.
- Section 2, sentence “It may be a basic material to keep sensitive activities for making a carrier ...”: this sentence is confusing. Do amend it.
- Section 3: why claim “ anthracis is not viable owing to its toxicity”? What is toxic here and what is not viable? Please check the logic of this sentence.
- Section 3: elaborate the sentence “Corynebacterium glutamicum may be suitable for membrane-anchored display of PgsBCA.”. The subsequent sentence is grammatical too.
- Figure 2: there are a lot of typo in this figure, e.g., crtric acid, suceinc, fumric, pyrurate. Please check the quality of adopted figure for this review.
- Figure 3: italicize genus name and gene names.
- Section 5.1: What do you mean by ‘simple screening and improvement’ here? Check the sentence again.
- Define abbreviation APFB.
- Tables: the authors insert general statement at the end of tables. Are these a note to the table? This style of writing is not conventional. You can include this general information in the main text instead, or else they may confuse the readers about the purpose of these statements.
- Italicize gene name ‘rocR’.
- Section 5.1: provide citations for sentence starting with ‘Many studies …’.
- Section 5.1: check the phrase ‘mutant deleted ggt in …’. Sounds grammatical. Similarly, check the phrase ‘mutants removing …’. Don’t start the sentence with “Because the gene”, as it is not a complete sentence.
- Section 5.1: when discussing the trends (e.g., higher yield, improvement in production), it is best to cite the values or provide them for a better comparison and explanation.
- Section 5.1: There is no proper explanation of the point ‘expression patterns’ raised in this section.
- Section 6: Biological differences or structural differences? Do check the working principles of those separation methods.
Author Response
Manuscript ID: Foods-1590602
Title: Recent Advances in Microbial Synthesis of Poly-γ-glutamic acid: A Review.
Authors: Danfeng Li, Lizhen Hou, Yaxin Gao, Zhiliang Tian, Bei Fan, Fengzhong Wang *, Shuying Li*
Foods
Dear Reviewers:
I only represent all co-authors, we are grateful to the editor and reviewers very much for their constructive comments on our manuscript (foods-1590602), and greatly thank you for considering the revised version of the manuscript. We have carefully considered all the reviewer’s comments and revised the manuscript accordingly. The revised sections are marked in red in the paper (Marked Manuscript: foods-1590602-v2). Detailed point-by-point responses are enclosed.
Attached please find the revised version, which we would like to submit for your kind consideration. And this is a response to your comments.
The major revisions are as follows:
- We have carefully revised the abstract and conclusion of manuscript.
- We have corrected errors in our manuscript.
- We have also revised and added to the highlights of manuscript more accurately according to reviewers’ comments.
- The list of references has been updated. We have referred to works published in recent years and adapted the relevant content of the manuscript according to the references.
We hope that the revised version is now suitable for publication in Foods: Probiotics, Prebiotics, Synbiotics, Postbiotics and Paraprobiotics – New Perspective for Functional Foods and Nutraceuticals, and we thank you for taking the time on this manuscript. The revised version has been approved by all co-authors of this paper.
Sincerely,
ShuYing Li, Associate Professor
Institute of food science and technology
Chinese Academy of Agricultural Sciences
Point-by-point responses to reviewers
Manuscript ID: Foods-1590602
Title: Recent Advances in Microbial Synthesis of Poly-γ-glutamic acid: A Review.
Reviewer #2:
Comment: This review focuses on the recent development in the aspect of gamma-PGA production by microorganism. The topic is more related to the fields of biotechnology and biochemistry, rather than food field; hence, the suitability of this review paper for publication in Food journal is questionable. In specific, l do not find a good relevance of this topic with the food element, except for a small section about gamma-PGA application in food industry. The authors claimed to provide a review on the 'recent advances, but most of the cited articles are prior to 2015.Pleaseensure that there is a sufficient development in this field before making a 'recent' review about the topic. In fact, the content covered by this review can be easily found in other available review papers (e.g.,10.1186/513068-016-0537-7). In general, the English grammar is acceptable but there are a number of awkward sentences and typos to be corrected. Other specific comments are given below:
Response: We are grateful for the positive comments from this reviewer. Our responses to the specific comments are set out below.
Comment 1. Q1: Correct grammars in phrases such as: “It always been”, “can efficiently enhanced’, “can efficiently convert into”, “have be tested”, “Ca2+”
Author Response: Thanks for the Reviewer’s very kind suggestions. They have all been corrected in the revised manuscript.
Comment 2. Q2: The citations for Figures 1-3 shows a general google website link. This is not the right way of acknowledging the source of citation. Do provide a proper citation in figure caption, and not in the main text.
Author Response: Thanks for the Reviewer’s very kind suggestions. We are very sorry for the confusion to you. We have misquoted the URL in the paper. We have updated the relevant citations.
Comment 3. Q3: Section 2, sentence “It may be a basic material to keep sensitive activities for making a carrier ...”: this sentence is confusing. Do amend it.
Author Response: Thanks for the Reviewer’s comment. This sentence has been amended in the revised manuscript in line 84.
Comment 4. Q4: Section 3: why claim “anthracis is not viable owing to its toxicity”? What is toxic here and what is not viable? Please check the logic of this sentence.
Author Response: Thanks for the Reviewer’s comment. B. anthracis is a pathogenic bacterium, therefore it is not viable for industrial applications. It has been corrected in the revised manuscript.
Comment 5. Q5: Section 3: elaborate the sentence “Corynebacterium glutamicum may be suitable for membrane-anchored display of PgsBCA.”. The subsequent sentence is grammatical too.
Author Response: Thanks for your very kind comments. We have carefully considered this sentence, it may confuse the readers in this place in manuscript, therefore we deleted this sentence, and later in the manuscript to elaborate other content about Corynebacterium glutamicum.
Comment 6. Q6: Figure 2: there are a lot of type in this figure, e.g., crtric acid, suceinc, fumric, pyrurate. Please check the quality of adopted figure for this review.
Author Response: We are very sorry for the above mistakes. It has been corrected in the figure.
Comment 7. Q7: Figure 3: italicize genus name and gene names.
Author Response: Thanks for the Reviewer’s nice suggestion. We have corrected errors in our manuscript, such as line 247.
Comment 8. Q8: Section 5.1: What do you mean by ‘simple screening and improvement’ here? Check the sentence again.
Author Response: Thanks for the Reviewer’s nice suggestion. Simple screening means isolated strains are screened by strain characteristics, such as thermal stability. Strain improvement means gene manipulation or modification for production strains. We checked the sentence and amended it.
Comment 9. Q9: Define abbreviation APFB.
Author Response: Thanks for the Reviewer’s comment. The abbreviation APFB is aerobic plant fibrous-bed bioreactor. Its definition has been added in the table 1.
Comment 10. Q10: Tables: the authors insert general statement at the end of tables. Are these a note to the table? This style of writing is not conventional. You can include this general information in the main text instead, or else they may confuse the readers about the purpose of these statements.
Author Response: Thanks for the Reviewer’s nice suggestion. I have revised the manuscript.
Comment 11. Q11: Italicize gene name ‘rocR’.
Author Response: Thanks for the Reviewer’s nice suggestion. We have corrected errors in our manuscript, such as line 247.
Comment 12. Q12: Section 5.1: provide citations for sentence starting with ‘Many studies …’.
Author Response: Thanks for the Reviewer’s very nice advice. We have amended this sentence and added some examples.
Comment 13. Q13: Section 5.1: check the phrase ‘mutant deleted ggt in …’. Sounds grammatical. Similarly, check the phrase ‘mutants removing …’. Don’t start the sentence with “Because the gene”, as it is not a complete sentence.
Author Response: Thanks for the Reviewer’s nice suggestion. We have rewritten the sentence in line 266.
Comment 14. Q14: Section 5.1: when discussing the trends (e.g., higher yield, improvement in production), it is best to cite the values or provide them for a better comparison and explanation.
Author Response: Thanks for the Reviewer’s very kind suggestions. In section 5, we have added more figures to be more convincing. And we referred to works published in recent years and adapted the relevant content of the manuscript according to the references.
Comment 15. Q15: Section 5.1: There is no proper explanation of the point ‘expression patterns’ raised in this section.
Author Response: Thanks for the Reviewer’s comment. We have added examples to illustrate constitutive and inducible expressions for explaining ‘expression patterns’ in line 298-304.
Comment 16. Q16: Section 6: Biological differences or structural differences? Do check the working principles of those separation methods.
Author Response: Thanks for the Reviewer’s very nice advice. The working principles of methods for purification mentioned is structural differences in this review. We have carefully revised this section.
Thank you very much for your comments and suggestions on our manuscript. We submitted the review for this special issue in Foods titled “Probiotics, Prebiotics, Synbiotics, Postbiotics and Paraprobiotics – New Perspective for Functional Foods and Nutraceuticals”. Bacillus species, as the probiotics, are the natural, dominant, and safely producers, and they have been used in fermented foods. γ-PGA is a natural, edible, biodegradable, biopolymer, which can be used in a wide range of food applications. Therefore, we think it is more related to the food field. We have tried our best to revise our manuscript in the light of the comments. Please find attached the revised version, which we would like to submit for your kind consideration.
Reviewer 3 Report
- The abstract should be more elaborative.
- Add more keywords
- More figures should be added to the manuscript
- Application in Other fields should be more elaborative with studies
Author Response
Manuscript ID: Foods-1590602
Title: Recent Advances in Microbial Synthesis of Poly-γ-glutamic acid: A Review.
Authors: Danfeng Li, Lizhen Hou, Yaxin Gao, Zhiliang Tian, Bei Fan, Fengzhong Wang *, Shuying Li*
Foods
Dear Reviewers:
I only represent all co-authors, we are grateful to the editor and reviewers very much for their constructive comments on our manuscript (foods-1590602), and greatly thank you for considering the revised version of the manuscript. We have carefully considered all the reviewer’s comments and revised the manuscript accordingly. The revised sections are marked in red in the paper (Marked Manuscript: foods-1590602-v2). Detailed point-by-point responses are enclosed.
Attached please find the revised version, which we would like to submit for your kind consideration. And this is a response to your comments.
The major revisions are as follows:
- We have carefully revised the abstract and conclusion of manuscript.
- We have corrected errors in our manuscript.
- We have also revised and added to the highlights of manuscript more accurately according to reviewers’ comments.
- The list of references has been updated. We have referred to works published in recent years and adapted the relevant content of the manuscript according to the references.
We hope that the revised version is now suitable for publication in Foods: Probiotics, Prebiotics, Synbiotics, Postbiotics and Paraprobiotics – New Perspective for Functional Foods and Nutraceuticals, and we thank you for taking the time on this manuscript. The revised version has been approved by all co-authors of this paper.
Sincerely,
ShuYing Li, Associate Professor
Institute of food science and technology
Chinese Academy of Agricultural Sciences
Point-by-point responses to reviewers
Manuscript ID: Foods-1590602
Title: Recent Advances in Microbial Synthesis of Poly-γ-glutamic acid: A Review.
Reviewer #3:
We are grateful for the positive comments from this reviewer. Our responses to the specific comments are set out below.
Comment 1. Q1: The abstract should be more elaborative.
Author Response: Thanks for the Reviewer’s for their kind advice. We have carefully revised the abstract of manuscript.
Comment 2. Q2: Add more keywords
Author Response: Thanks for the Reviewer’s very kind suggestions. We have added two keywords: Bacillus species and industrial application.
Comment 3. Q3: More figures should be added to the manuscript
Author Response: Thanks for the Reviewer’s very kind suggestions. In section 5, we have added more figures to be more convincing. And we referred to works published in recent years and adapted the relevant content of the manuscript according to the references.
Comment 4. Q4: Application in Other fields should be more elaborative with studies.
Author Response: Thanks for the Reviewer’s for their kind advice. We submitted the review for this special issue in Foods titled “Probiotics, Prebiotics, Synbiotics, Postbiotics and Paraprobiotics – New Perspective for Functional Foods and Nutraceuticals”. We feel it is more important to focus on food field. Therefore, we give the food field more thought and studies, application in other fields were had a brief study.
Thank you very much for your comments and suggestions on our manuscript. We have tried our best to revise our manuscript in the light of the comments. Please find attached the revised version, which we would like to submit for your kind consideration.
Reviewer 4 Report
This is an interesting review, and the authors have collected the research information on microbial biosynthesis, together with industrial applications of γ-PGA. The paper is generally well structured and easy to follow. However, in my opinion, some aspects mentioned in the paper are not described clearly, and the author has misrepresented particular findings. Also, the paper needs extensive revision for language and grammar. Given these shortcomings the manuscript requires major revisions.

Author Response
Manuscript ID: Foods-1590602
Title: Recent Advances in Microbial Synthesis of Poly-γ-glutamic acid: A Review.
Authors: Danfeng Li, Lizhen Hou, Yaxin Gao, Zhiliang Tian, Bei Fan, Fengzhong Wang *, Shuying Li*
Foods
Dear Reviewers:
I only represent all co-authors, we are grateful to the editor and reviewers very much for their constructive comments on our manuscript (foods-1590602), and greatly thank you for considering the revised version of the manuscript. We have carefully considered all the reviewer’s comments and revised the manuscript accordingly. The revised sections are marked in red in the paper (Marked Manuscript: foods-1590602-v2).
Attached please find the revised version, which we would like to submit for your kind consideration. And this is a response to your comments.
The major revisions are as follows:
- We have carefully revised the abstract and conclusion of manuscript.
- We have corrected errors in our manuscript.
- We have also revised and added to the highlights of manuscript more accurately according to reviewers’ comments.
- The list of references has been updated. We have referred to works published in recent years and adapted the relevant content of the manuscript according to the references.
We hope that the revised version is now suitable for publication in Foods: Probiotics, Prebiotics, Synbiotics, Postbiotics and Paraprobiotics – New Perspective for Functional Foods and Nutraceuticals, and we thank you for taking the time on this manuscript. The revised version has been approved by all co-authors of this paper.
Sincerely,
ShuYing Li, Associate Professor
Institute of food science and technology
Chinese Academy of Agricultural Sciences
Reviewer #4:
Comment: This is an interesting review, and the authors have collected the research information on microbial biosynthesis, together with industrial applications of y-PGA. The paper is generally well structured and easy to follow. However, in my opinion, some aspects mentioned in the paper are not described clearly, and the author has misrepresented particular findings. Also, the paper needs extensive revision for language and grammar. Given these shortcomings the manuscript requires major revisions.
Response: We are grateful for the positive comments from this reviewer. Our responses to the specific comments have been fully reflected in the revised manuscript. I am sorry I cannot answer you point by point because of the ‘pdf version’.
Thank you very much for your comments and suggestions on our manuscript. We have tried our best to revise our manuscript in the light of the comments. Please find attached the revised version, which we would like to submit for your kind consideration.
Round 2
Reviewer 2 Report
All the comments have been addressed. Please check the English grammar again before publishing this paper.
Reviewer 4 Report
In general, the revised version of the manuscript appears to be good, as the authors have made the necessary changes and corrections. Particularly, I find the manuscript now possesses a concise and readable writing style, giving a clearer explanation regarding biological synthesis of γ-PGA. As far as I can see, the manuscript looks ready for publication.